# Color Doppler Ultrasound Improves Machine Learning Diagnosis of Breast Cancer

**DOI:** 10.3390/diagnostics10090631

**Published:** 2020-08-25

**Authors:** Afaf F. Moustafa, Theodore W. Cary, Laith R. Sultan, Susan M. Schultz, Emily F. Conant, Santosh S. Venkatesh, Chandra M. Sehgal

**Affiliations:** 1New York Medical College, Valhalla, NY 10595, USA; amoustaf@student.nymc.edu; 2Department of Radiology, University of Pennsylvania, Philadelphia, PA 19104, USA; lsultan@pennmedicine.upenn.edu (L.R.S.); Susan.Schultz@pennmedicine.upenn.edu (S.M.S.); emily.conant@pennmedicine.upenn.edu (E.F.C.); Chandra.Sehgal@pennmedicine.upenn.edu (C.M.S.); 3Department of Electrical and Systems Engineering, University of Pennsylvania, Philadelphia, PA 19104, USA; venkates@seas.upenn.edu

**Keywords:** ultrasound, color Doppler, radiomics, breast cancer, machine learning

## Abstract

Color Doppler is used in the clinic for visually assessing the vascularity of breast masses on ultrasound, to aid in determining the likelihood of malignancy. In this study, quantitative color Doppler radiomics features were algorithmically extracted from breast sonograms for machine learning, producing a diagnostic model for breast cancer with higher performance than models based on grayscale and clinical category from the Breast Imaging Reporting and Data System for ultrasound (BI-RADS_US_). Ultrasound images of 159 solid masses were analyzed. Algorithms extracted nine grayscale features and two color Doppler features. These features, along with patient age and BI-RADS_US_ category, were used to train an AdaBoost ensemble classifier. Though training on computer-extracted grayscale features and color Doppler features each significantly increased performance over that of models trained on clinical features, as measured by the area under the receiver operating characteristic (ROC) curve, training on both color Doppler and grayscale further increased the ROC area, from 0.925 ± 0.022 to 0.958 ± 0.013. Pruning low-confidence cases at 20% improved this to 0.986 ± 0.007 with 100% sensitivity, whereas 64% of the cases had to be pruned to reach this performance without color Doppler. Fewer borderline diagnoses and higher ROC performance were both achieved for diagnostic models of breast cancer on ultrasound by machine learning on color Doppler features.

## 1. Introduction

Breast cancer is diagnosed in one in eight women, making up 30% of all new cancer cases in women [1]. In 2018 alone, over two million cases of breast cancer were diagnosed, and over 600,000 deaths related to breast cancer occurred worldwide [2]. Breast cancer screenings and early detection are vital for reducing the mortality rate. Mammography is currently the primary mode of screening for average-risk women, yet it can miss up to 20% of cases [3]. Ultrasound (US) is typically included as a diagnostic follow-up measure, only after a clinical examination and mammography [3,4]. Ultrasound is also increasingly used as a supplemental screening tool, particularly in women with dense breasts, where mammographic sensitivity is limited. In order for ultrasound screening to be widely implemented, both the sensitivity and specificity of sonography must be improved [5]. Recent studies have estimated the sensitivity of US to be 75%, 81%, and 83%, depending on patient population [6,7,8]. Currently, the specificity of US in breast cancer diagnosis is also low, and between 70% and 85% of breast biopsies prove to be benign [9,10]. These unnecessary biopsies result in both emotional and physical trauma to the patient as well as in socioeconomic costs on a wider scale.

Information from color Doppler (CD) images can improve breast US diagnostic performance. CD allows for the clinical assessment of blood flow velocity, as well as of the degree of vascularity, within and around breast masses [11]. Hypervascularity, though not pathognomonic of malignant lesions, is strongly correlated with malignancy. Through a series of intracellular pathways, tumor cells secrete angiogenin, which directly activates neoangiogenesis [12,13]. Angiogenin levels and the corresponding neoangiogenesis are correlated with clinical progression in the metastatic development of breast cancer [13]. Neoangiogenesis in breast lesions is also associated with increased speed of tumor growth and metastasis [14]. Although the assessment of vascularity can therefore improve breast cancer diagnosis using ultrasound [5,15,16,17,18,19], Doppler imaging is still underused in clinical applications. Since BI-RAD_US_ grading does take Doppler images into account, there is scope for formalizing and quantifying Doppler assessment for improving breast mass diagnosis with radiomics. 

The sensitivity and specificity of breast US can be improved through machine learning, which acts as a second reader to assist the radiologist in making a diagnosis [5,9,10,20]. Machine learning algorithms learn characteristics of known cases in order to build models for diagnosing patients with unknown disease states. Logistic regression and naïve Bayes are two proven classifiers with different learning strategies that nevertheless strongly agree on sonographic breast mass cancer diagnosis on a case-by-case basis [21]. The two classifiers have been combined into a higher-performing model trained on grayscale (GS) features alone [10] using serial adaptive boosting (AdaBoost [22]). Instead of boosting a homogenous weak learner as in canonical AdaBoost, the method boosts heterogeneous classifiers, such as naïve Bayes and logistic regression, that are already known to both perform well and to even closely agree on cases [10,20,21,23].

This study algorithmically extracted numerical CD features from US images for machine learning, in addition to grayscale numerical features. All these computer-generated features were used along with age and the category from the Breast Imaging Reporting and Data System for iltrasound (BI-RADS_US_) used to train models [24]. The benign–malignant class imbalance in the training data was addressed using the synthetic minority oversampling technique (SMOTE) [25]. The motivation for the research was to determine whether the addition of CD features could significantly improve performance beyond what was possible using grayscale features and BI-RADS_US_. In order to investigate the role of CD in improving the diagnostic performance of machine learning, after an overview and exploratory analysis of the images and feature sets used for training, machine learning was performed using the UltraBoost computer-aided diagnosis (CAD) algorithm. The interface was improved to accept any number of constituent classifiers available in the Weka open-source platform. The algorithm generated models for differentiating malignant from benign masses on breast ultrasound, which were compared for diagnostic performance.

## 2. Materials and Methods

### 2.1. Patient and Image Acquisition

Ultrasound images of 159 solid masses from 156 patients were analyzed on a retrospective database with approval from the institutional review committee, (IRB registration 824555, approved on 2 April 2019). The images were obtained from a Philips HDI 5000 Ultrasound System (Philips Healthcare, Bothell, WA, USA), operated by a registered diagnostic medical sonographer (RDMS) with over 20 years of experience. Based on a separate mammographic examination, each patient in the study had also undergone a biopsy. The biopsy outcomes, 95 benign and 64 malignant, were the reference standard for training and testing the models. This work built upon earlier research that used only grayscale B-Mode features and clinical features without any consideration of color Doppler [10,21]. The images were taken from the same database used in the earlier retrospective studies, but they comprised only a subset of more difficult cases that also had color Doppler in addition to B-Mode, since color Doppler was only recorded for masses that had been visually assessed to be problematic. 

### 2.2. Feature Extraction

#### 2.2.1. BI-RADS_US_

Clinically determined BI-RADS_US_ categories (0 to 5) were input as features to train the classifiers, with the following distribution, in category:count format: 0:12, 1:6, 2:9, 3:4, 4:66, 5:38. Most cases (104, 65%) were Category 4 or 5, and 24 cases (15%) were missing BI-RADS_US_ classifications. In the ensemble model, BI-RADS_US_ was only relevant to the naïve Bayes constituent classifier, which naturally trains on discrete class attributes and naturally handles missing records by omitting them from model probability calculations. There was less BI-RADS_US_ information available than in prior studies, which also trained on a detailed BI-RADS_US_ checklist of present or absent visually assessed diagnostic features. Only the coarse BI-RADS_US_ category was available for this research, corresponding more closely to what is typically available in a clinical database. In summary, the database for this study drew from a subset of more difficult images, with less clinical information available from experts, but the study expands the feature space to color Doppler. 

#### 2.2.2. Age

The ages for all but three patients were obtained from pathology reports. For these three patients, the mean age of all the other patients was used (50.2 ± 13.8; range, 21 to 84 years).

#### 2.2.3. Grayscale Features 

In order to calculate GS features, regions of interest (ROIs) were manually drawn using a custom application written in the IDL (Interactive Data Language) programming language (version 8.5; Harris Geospatial, Broomfield, CO, USA). One to ten ROIs were extracted per mass, with a mean of 3.16 ROIs. Each numeric feature was then averaged across all the ROIs per mass. These nine features were motivated by the conventional visual features used in the clinic to assess lesions on ultrasound, described by Stavros et al. [26]. They include the angular variation at the margin (AVM), angular variation of the interior (AVI), brightness difference (BD), margin sharpness (MS), tortuosity, depth-to-width ratio (DWR), axis ratio (AR), radius variation (RV), and ellipse-normalized skeleton (ENS) [9,14,23]. The formulas for deriving the features are given in the literature [9,10] and in the Appendix A
Table A1.

#### 2.2.4. Color Doppler Features

Two CD features were used to measure vascularity within each manually drawn ROI: the vascular fractional area (VFA) and blood flow velocity index (VI). Though the features are used in this study for machine learning, they were originally designed as measurements for assessing antivascular treatments [27]. The color scale was set by dividing directional flow into 100 equal levels from the color-bar on the Doppler image, between 0 and the maximum velocity on the bar. All colored pixels within the ROI were then assigned a velocity index mapped from the color-bar gradient. The number of colored pixels within the ROI, or the number of pixels with flow, and their velocity indices were then used to measure the VFA and VI, as described in both in the Appendix A
Table A1and in the literature [5,14].

### 2.3. Machine Learning

The UltraBoost classifier to adaptively boost heterogeneous learners was coded in Java and used to train the models in Weka 3.8, assigning each case a probability of malignancy for diagnosis [28,29]. “Ultra” refers to its original purpose for diagnosing ultrasound images using a two-learner-boosted ensemble, although this research generalized the code and interface to accept any number of constituent classifiers from those available as open source in Weka. Although canonical AdaBoost boosts weak homogeneous learners, the motivation for boosting already-strong heterogeneous classifiers is that if one classifier already has very high performance, the optimal classification strategy is likely close to the already-strong strategy, so it is boosted with a learner that provides a second opinion. Previous studies found that naïve Bayes and logistic regression both performed well on the breast mass classification problem and, then, that they could be boosted even though their predictions were close case by case [10,20,21,23]. Though stacking is usually used instead of boosting with heterogeneous learners, the close agreement by case between our constituent learners makes them poor candidates for stacking. When boosting, though the second learner has problems with the same cases, it learns their attributes using different logic, so the final ensemble still benefits when the first classifier weights the problematic cases so that they are examined more closely by the second classifier. The boost is performed only once between classifiers rather than iteratively in as many stages as it takes to minimize some objective function, as in traditional AdaBoost, since the optimization is less well behaved and more likely to over-fit when using many heterogeneous learners. A practitioner’s regularization on attributes was performed by filtering numerical GS, CD, and age attributes to the constituent logistic regression classifier, while the categorical BI-RADS_US_ attribute was filtered earlier to the constituent naïve Bayes classifier. 

The class attribute predicted by the model was diagnosis—1 for malignant and 0 for benign—so the output probabilities of malignancy ranged from 0 to 1. The model was constructed using leave-one-out cross validation (LOOCV), in which *n* − 1 samples (*n* = number of cases) were trained to predict the probability of malignancy for the remaining *n*-th sample, repeated *n* times to predict every case, for a total of 159 folds. LOOCV ensures that test cases are always left out of training while training still occurs on the most possible data, which is necessary since the dataset is too small for a larger sample-split validation supporting 12 features. The probability of malignancy for each case was then compared with actual biopsy results receiver operating characteristic curve (ROC) analysis using MedCalc (version 19.0.5, MedCalc Software Ltd., Ostend, Belgium). MedCalc generated the area under the ROC curve (AUC), as well as the sensitivity and specificity at the Youden Index for each ROC curve, to evaluate the diagnostic performance of the model. Though Youden values were calculated for reference, sensitivity is more important for detecting cancer than the Youden-index maximization of the sum of sensitivity and specificity [30]; therefore, the models’ performances at high-sensitivity thresholds of 95% and 98% are also reported.

### 2.4. Class Imbalance Correction Using SMOTE

The same modeling and analysis above was performed after correcting the database for the benign–malignant class imbalance using the synthetic minority oversampling technique (SMOTE) in Weka [25,28,29]. Of the 159 cases in the database, 95 were benign and 64, malignant. Without correcting for the sampling bias, models trained on the data would tend towards predicting cases as benign to minimize error and achieve the highest ROC performance. Although this performance may be generalizable to future cases as long as the clinic continues to acquire benign and malignant cases in the same proportion from the same population, there are, of course, higher real-world costs to missing malignant diagnoses, that are not captured in ROC metrics. These costs are difficult to exactly calculate beforehand, so instead of using cost-sensitive learning or cost-based thresholding on the ROC curves, a principled oversampling technique is used, SMOTE, which constructs synthetic cases to balance the dataset. Note that although the models were trained using the synthetic examples, their performance was calculated by applying the models using LOOCV only on the original real-world cases without the synthetic examples, to prevent training leakage. 

### 2.5. Pruning by Drop Rate

The performance of each model was also compared when low-confidence cases were pruned. In classifying each mass as benign or malignant, the UltraBoost classifier outputs a probability *p* where 0 ≤ *p* ≤ 1. Values near 0 represent high confidence that the case is benign, while values near 1 represent high confidence that the case is malignant. The threshold for diagnosis was set at 0.5, with 0.5 being the lowest possible probability with a malignant prediction. As the classifier becomes less confident in a case, the probability output approaches the 0.5 threshold. The AUC was assessed at different drop rates by pruning low-confidence cases in a symmetric range around *p* = 0.5 comprising a set percentage of all cases. If multiple cases had the same low-confidence probability, the cases were pruned together. The AUC for the remaining unpruned cases was then calculated.

## 3. Results

### 3.1. Patient and Image Characteristics

Of the 64 masses with malignant histology, 47 (73%) were infiltrating ductal carcinomas; 6 (9%), invasive lobular carcinoma; 4 (6%), ductal carcinoma in situ; and 7 (11%) were unclassified lesions. Of the 95 that were histologically benign, 44 (46%) were fibroadenomas; 13 (14%), miscellaneous fibrocystic changes; 4 (4%), ductal hyperplasias; 4 (4%), sclerosing adenosis; and 30 (32%) were otherwise-benign lesions without atypia. The age range of the patients was 21.2 to 83.9 years, with a mean and standard deviation of 50.6 ± 13.8. 

Figure 1 provides examples of GS and CD US images. The GS ultrasound image of a 47-year-old woman (left panel, Figure 1a) shows a hypoechoic mass. No vascularization is shown in the mass’s interior, and instead, the vascularization is only found outside the ROI (right panel, Figure 1a). The mass was clinically assessed to be BI-RADS_US_ Category 4, suspicious. The computer-extracted quantitative features for the mass were 3.85 for AVI, 3.85 for AVM, 12.9 for BD, 75.7 for MS, 1.45 for AR, 0.877 for DWR, 0.19 for RV, 0.17 for ENS, 1.25 for tortuosity, 0 for VI, and 0 for VFA. While the algorithm predicted that this mass was malignant when CD features were not used, adding CD features allowed the model to correctly predict the mass as benign. 

The GS ultrasound imaging of a 38-year-old woman shows another hypoechoic mass (left panel, Figure 1b). Unlike the case shown in Figure 1a, the mass is highly vascular (right panel, Figure 1b). The BI-RADS_US_ category of the mass was also clinically assessed to be Category 4, suspicious. The computer-extracted quantitative features for the mass were 5.39 for AVI, 3.09 for AVM, 13.7 for BD, 78.5 for MS, 1.40 for AR, 1.04 for DWR, 0.18 for RV, 0.17 for ENS, 1.25 for tortuosity, 1.79 for VI, and 17.1 for VFA. The algorithm incorrectly predicted this mass to be benign when CD features were not used. However, when the model incorporated CD features, the mass was correctly predicted to be malignant.

### 3.2. Feature Statistics and Effect Size

Nine GS features and two CD features were extracted by algorithms as described in Section 2.2 Feature Extraction. The BI-RADS_US_ category and patient age were obtained from pathology reports. Although machine learning models can capture the predictive power of all the attributes together that were used to train the classifiers, preliminary exploratory data analysis was also performed on the features using descriptive statistics and the Mann–Whitney U test. This is exploratory analysis, not results from running the final machine learning algorithm. Table 1 displays the mean and standard deviation for each feature within the benign and malignant groups, along with the *p* values for the difference of each feature between the two groups. Only the following features taken by themselves demonstrated statistically significant differences (*p* < 0.05) between benign and malignant masses: BI-RADS_US_, age, BD, MS, DWR, ENS, tortuosity, VI, and VFA. If the Bonferroni correction is applied for the twelve features, lowering the significance threshold to 0.00417, then the ENS, axis ratio, and tortuosity, all shape features, are no longer significant. Both CD features, VI and VFA, were statistically highly significant regardless of Bonferroni correction (*p* < 0.001). The use of Bonferroni correction may not be appropriate for detecting cancer, since it increases the chances of false negatives [31]. The U tests are not the final model resulting from this research, only exploratory data analysis; the models that capture interactions between features were generated by machine learning, described below in Section 3.3 Performance of Each Diagnostic Model, and the effect size of the numeric features from logistic regression was used to quantify their relative predictive power.

The effect sizes of each numeric feature from logistic regression are listed in Table 1. The effect size used is the standardized directionless odds ratio of the highest-AUC unpruned model (0.958 ± 0.013, Table 2), trained on all features and all cases. A “+” in the effect size denotes that an increase in the numeric feature corresponds to higher odds of malignancy. Feature values were standardized for a 0 mean and unit variance before training so that their effect sizes could be compared. The reported effect size is a ratio that measures how many times greater the odds of diagnosis would become if the standardized numeric feature were to increase by one standard deviation. For instance, the effect size of 5.78+ for the age feature means that an increase of 1 standard deviation in age, or 13.3 years, corresponds to an increase in malignancy odds by a factor of 5.78, assuming nothing else changes. Brightness difference (BD) has no + after the effect size, so it is anti-correlated with malignancy, and an increase of one standard deviation increases the odds of the case being *benign* by a factor of 12.0; this is to be expected since a large brightness difference at the margin indicates that the lesion is well defined, typical of a benign mass. Effect sizes are useful for comparing the relative predictive power of the numeric features. Age (5.78+ effect size) and angular variation in margin (10.3+) are highly correlated with malignancy; notably, the two Doppler features, VI (2.63+) and VFA (2.12+), have nearly the same effect size on malignancy odds as the traditional clinical shape feature of the depth-to-width ratio (DWR, 2.38+).

### 3.3. Performance of Each Diagnostic Model

Table 2 summarizes the performance of the models trained on different feature sets, as measured by the AUC, with and without correcting for class bias using SMOTE, with the highest-performing model being the one trained on all features: BI-RADS_US_, age, CD, and GS, with an AUC of 0.958 ± 0.013 after SMOTE correction. In general, as more features are added to the baseline BI-RADS_US_-and-age set, the AUC performance improves, as shown in Figure 2. Performance is, of course, the lowest when only considering BI-RADS_US_, at 0.664 ± 0.052, or 0.770 ± 0.051 (SMOTE). When age is added, the performance increases to 0.854 ± 0.031. The performance continues to increase once GS and CD features are each added: BI-RADS_US_, age, and GS (all features except CD) performed second best (0.900 ± 0.024, 0.925 ± 0.022 SMOTE), followed by the BI-RADS_US_, age, and CD groups (0.891 ± 0.026, 0.901 ± 0.025 SMOTE). In Table 3, the performance of all the features including CD is statistically significantly better than that of all the features without CD (*p* = 0.0161, 0.0352 SMOTE).

At the Youden index, the model trained on the group containing all the features (BI-RADS_US_, age, nine GS features, and two CD features) was the most sensitive (Se = 96.9%), followed by BI-RADS_US_, age, and GS (Se = 85.9%), which was tied with BI-RADS_US_ and age alone, and then the feature set of BI-RADS_US_, age, and CD (Se = 73.4%) (Table 2). At a high-sensitivity operating threshold of 98.4%, the all-inclusive feature set including both GS and CD had the highest specificity of 76.8%, much higher than that of GS (32.6%) or CD (26.3%) alone. When the sensitivity was fixed at 95.3%, the specificity of the GS+CD feature set was 85.3%, again, higher than that of GS (67.4%) or CD (60.0%). Adding CD to the features increases the model specificity when operating at the high-sensitivity thresholds necessary for cancer detection.

### 3.4. Performance of Models Pruned by Drop Rate

A larger increase in performance can be seen once low-confidence cases are pruned. Cases where the classifier has the lowest confidence can be singled out for further testing. Table 4 and Figure 3 illustrate the effect of using a pruning threshold on the two feature sets, one with CD features and one without. As seen in Figure 3, the curve including CD features eventually reaches a plateau, where the benefit of increased performance is outweighed by the larger drop rate. A 20% drop (32 cases: 18 malignant, 14 benign) for the CD feature set significantly increases performance, with an AUC of 0.986 ± 0.007. Meanwhile, more than triple the number of cases, 64% (101 cases: 36 malignant, 65 benign), must be pruned in order to reach the equivalent 0.986 AUC performance without CD features.

## 4. Discussion

Radiomics, the diagnosis of medical images through the automated extraction and analysis of predictive quantitative features, has been increasingly used in medical imaging, especially X-ray tomography and tomosynthesis, computer tomography (CT), positron emission tomography (PET), and magnetic resonance imaging (MRI) [32]. Most breast cancer radiomics studies are on MRI or mammography, and relatively few use ultrasound features [33,34,35,36]. The results show a statistically significant improvement in performance by combining both GS and CD ultrasound features with clinically available features, BI-RADS_US_ and patient age, for training a boosted heterogeneous ensemble machine learning algorithm, UltraBoost, especially when GS and CD features were both added in combination to the clinical features. 

The features analyzed by UltraBoost included BI-RADS_US_, age, GS, and CD subsets. BI-RADS_US_ and age were considered the baseline set. GS and CD features were added in order to improve upon baseline performance. Though the study aimed to determine the feasibility of diagnosing lesions utilizing these computer-extracted features, they are not intended to replace a radiologist’s readings, so BI-RADS_US_ classification was retained as a feature. Additionally, a recent study by Watanabe et al. found that differences in US between benign and malignant breast masses, especially in the vascularity revealed by CD, can depend on the patient’s age, supporting the selection of age as a feature [15].

The GS features described above characterize the morphology (AR, DWR, RV, ENS, and tortuosity) and intensity (AVM, AVI, BD, and MS) of the lesion. As shown in Table 1, not all of these features taken separately showed statistically significant differences between the benign and malignant masses, according to the Mann–Whitney U tests performed in preliminary analysis. However, each of these features improved the performance of the machine learning models. With the noted exception of AVM, the features with the smallest effect sizes all had smaller *p* values, so it was unsurprising that adding these GS features to the clinical BI-RADS_US_ and age dataset significantly improved performance (Table 2; Figure 2). There was a highly statistically significant difference in machine learning AUC performance with and without GS features (*p* = 0.00860, 0.0116 SMOTE; Table 3). The sensitivity and specificity at the Youden Index, as well as the specificity at fixed sensitivities of 95.3% and 98.4%, all increased as well.

In order to further improve upon this performance, CD features were also added to train the model. Though utilizing CD for differentiating benign and malignant masses is not currently clinical standard practice, there is much evidence of its potential benefit. Previous studies have shown that CD improves the sensitivity and specificity of cancer diagnosis when utilized with GS US and clinical data [5,15,16,17,18,19]. This is thought to be because CD provides unique information on the direction and velocity of blood flow while also minimizing color artifacts. Information on the directionality and velocity of blood flow and vascular coverage were captured by the CD features in this study, accounting for their contribution to the improved diagnostic performance. Previous research demonstrates how higher VFA and VI might be predictive of malignant diagnosis: an increase in VFA would indicate an increase in vascularity. Similarly, the velocity index VI is predictive because malignant tumors are more likely to have formed arteriovenous shunts and to contain thin-walled vessels, both of which result in high-pulsatility, high-velocity flow [16].

Both of the CD features, VFA and VI, were, in fact, predictive of malignancy, consistent with the pathology described above. Each feature showed a highly statistically significant difference between the benign and malignant groups (*p* < 0.0001) in the preliminary exploratory analysis, and, during machine learning, adding CD features to the set of BI-RADS_US_, age, and GS significantly improved performance (*p* = 0.0161, 0.0352 SMOTE). Adding CD features to only clinical features (BI-RADS_US_ and age) also improved performance significantly (*p* = 0.0320) but not when the model was corrected for class imbalance (*p* = 0.136). The paired *p* value performance of baseline+CD features was similar to that of baseline+GS features (*p* = 0.760, 0.375 SMOTE), where the non-significant difference in *p* values indicates that both GS and CD added similarly helpful new information to the baseline clinical BI-RADS_US_+age model, regardless of SMOTE class imbalance correction. Although when they are independently added to the baseline, the performance increase is similar, in combination, GS and CD features significantly improve diagnostic performance. The most important result is the highly significant difference in performance with and without CD features, with the AUC increasing from 0.925 ± 0.022 to 0.958 ± 0.013 (Table 2). The sensitivity and specificity at the Youden Index, as well as the specificity at fixed sensitivities of 95.3% and 98.4%, improved as well.

While the ROC performance is above 0.95 AUC when utilizing all these features and correcting for class imbalance (AUC = 0.958 ± 0.022), the classifier is not able to diagnose all cases with high confidence. Pruning low-confidence cases increased the performance of the model, as shown in Figure 3. Pruning improved performance regardless of whether CD features were used to train the model, unsurprisingly since the difficult cases were being removed, but the important result is that this improvement was significantly greater when CD features were included. In the CD feature set, pruning 20% of the cases (32 cases) increased the AUC to 0.986 ± 0.007, with a sensitivity of 100%. By contrast, when CD features were not included, pruning 20% of the cases only resulted in an AUC of 0.944 ± 0.022. More than triple the number of cases must be pruned to match the performance of the pruned CD model: 0.986 AUC was reached only when 64% of the cases (101 cases) were pruned.

These pruned cases are the cases with low confidence in diagnosis. They would be sent for further evaluation by other diagnostic methods. Upon pruning 20% of the cases from the CD model, the sensitivity improved from 96.7% to 100%, which is why 20% is suggested as a drop rate; no cancers were missed at this level. Though the pruned model requires further workup for the dropped cases, it should reduce the health risks and mortality associated with false negatives.

The study has limitations. Firstly, in order to generate GS and CD features, manual delineation of the ROI had to be performed on multiple images for every case. This is a time-consuming process that must be completed by a trained observer, but manual ROI selection is a limitation of many CAD studies. Well-selected ROIs improve performance by facilitating the radiomic extraction of representative GS and CD features. Because ROI delineation requires experts’ time and effort, and because of privacy concerns, there are no large publicly available labeled color Doppler ultrasound image databases, but the 159 cases here are sufficient for determining that color Doppler features improve diagnostic performance, a result that should generalize to other scanners over more cases.

As with most clinically obtained real-world data, the class distribution of the cases was unbalanced, and there were missing data. The class bias was addressed using SMOTE to create synthetic cases for training the model. Three cases were missing age, and 15% of the cases were not given a BI-RADS_US_ category by a clinician. These missing data most likely decreased model performance; the models achieved the reported performance even though they were trained with less information. 

The majority of the masses in this study fell into three categories: fibroadenomas, infiltrating ductal carcinomas, and in situ carcinomas, which made up 95, or 60%, of the 159 masses. Previous studies have found that CD is less helpful in diagnosing certain types of breast lesions, including mucinous carcinomas, in situ carcinomas, infiltrating ductal carcinomas smaller than 9 mm, and papillary lesions, as they tend to lack the vascularity characteristic of malignant masses. CD may also lead to false-positive malignant diagnosis in fibroadenomas and papillomas, as they can present as hypervascularized [12]. If a larger imaging database were available, the analysis could be broken down by lesion histotype to determine which could be best diagnosed by machine learning using CD.

This research created models to diagnose breast cancer on ultrasound using computer-extracted CD and GS features as well as the clinically assessed feature of the BI-RADS_US_ category. Training on the two CD features, VFA and VI, improved the performance of the models, especially under the drop-rate pruning of indeterminate cases. In recent years, the improved resolution of CD has allowed heightened detection of blood flow in both benign and malignant masses [12,15,37]. This advance in CD technology could improve both the sensitivity and specificity of breast lesion diagnosis by machine learning. Future studies may expand to other, more nuanced CD features. For example, it has been hypothesized that penetrating vessels are more likely to be present in malignant tumors. Malignant tumors are also more likely to have central vascularization, as well as branching, disordered, and intra-tumoral vessels [12]. Future studies on larger databases could include these features of vascularization to determine if they also increase the diagnostic performance of machine learning. 

## Figures and Tables

**Figure 1 diagnostics-10-00631-f001:**
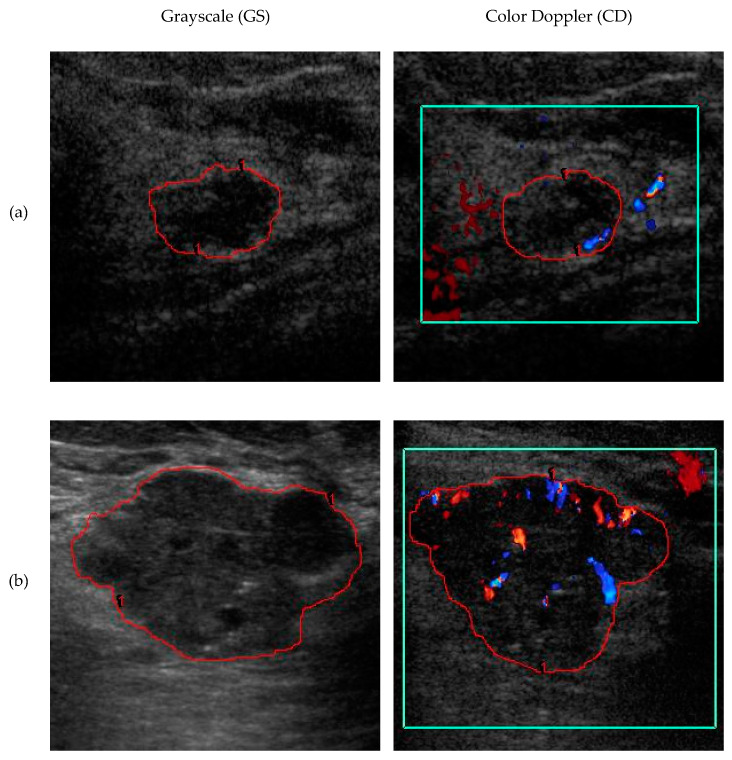
GS and CD ultrasound images of two breast masses. Each row contains one mass. The red outlines indicate the manually drawn ROI. The ROI is slightly different between CD and GS because the scanning plane changed. The green box is the Doppler box defined by the sonographer for displaying CD signal. (**a**) Benign mass, BI-RADS_US_ Category 4, 47-year-old woman. (**b**) Malignant mass, BI-RADS_US_ Category 4, 38-year-old woman. In both cases, the predictive model trained without CD features diagnosed the mass incorrectly, but with CD features the model’s diagnosis was correct. (GS = grayscale; CD = color Doppler; BI-RADS_US_ = Breast Imaging Reporting and Data System for ultrasound).

**Figure 2 diagnostics-10-00631-f002:**
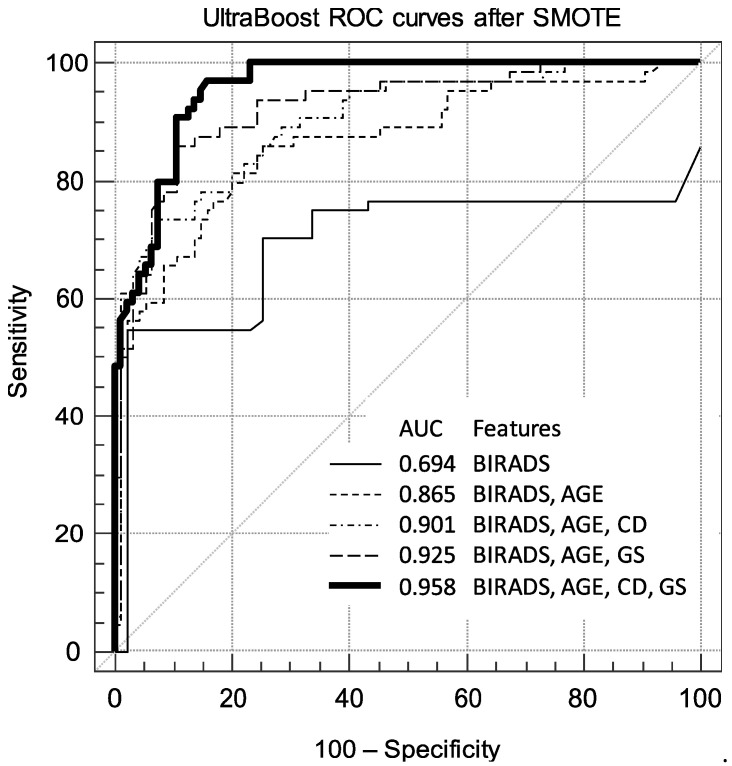
ROC curves for UltraBoost model with and without GS and CD features. Using CD features improves the diagnostic performance. Adding CD features to BI-RADS_US_ and age increased performance from AUC = 0.865 ± 0.031 to 0.901 ± 0.025; adding CD features to the above and GS features increased the AUC from 0.925 ± 0.022 to 0.958 ± 0.013. (GS = grayscale; CD = color Doppler; AUC = area under the ROC curve; ROC = receiver operating characteristic; BI-RADS_US_ = Breast Imaging Data and Reporting System for ultrasound).

**Figure 3 diagnostics-10-00631-f003:**
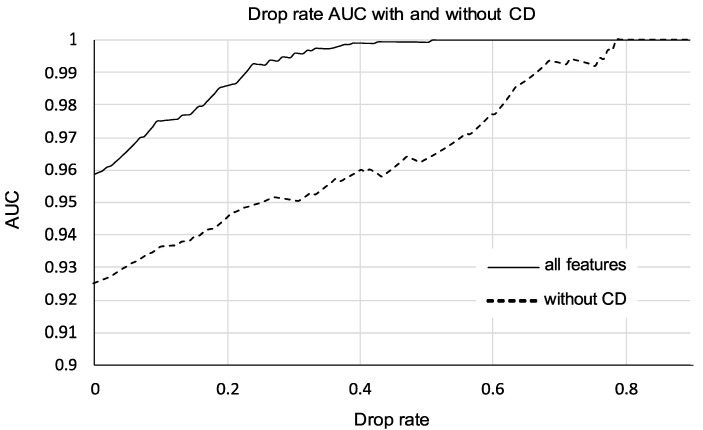
Drop rate curve with and without CD features. Using CD features reduces proportion of indeterminate cases. Pruning such cases at a 20% drop rate increased performance from AUC = 0.958 ± 0.022 to AUC = 0.986 ± 0.007 with 100% sensitivity. (CD = Color Doppler; AUC = Area under the receiver operating characteristic curve).

**Table 1 diagnostics-10-00631-t001:** Numeric feature distribution, significance, and effect size.

Feature	Benign	Malignant	U-Test *p* Value	Effect Size
Age	46.2 ± 13.3	57.1 ± 11.5	**<0.0001**	5.78+
Angular interior (AVI)	4.18 ± 1.66	4.24 ± 1.84	0.4441	2.97
Angular margin (AVM)	3.35 ± 1.13	3.60 ± 1.46	0.2261	10.3+
Bright difference (BD)	17.2 ± 7.2	11.0 ± 6.2	**<0.0001**	12.00
Margin sharpness (MS)	82.8 ± 8.1	74.0 ± 10.2	**<0.0001**	1.97
Axis ratio (AR)	1.74 ± 0.46	1.60 ± 0.37	**0.0308**	5.89+
Depth–width ratio (DWR)	0.71 ± 0.19	0.86 ± 0.27	**0.0002**	2.38+
Radius variation (RV)	0.21 ± 0.07	0.19 ± 0.07	0.1261	2.36
Skeleton (ENS)	0.14 ± 0.02	0.15 ± 0.03	**0.0113**	1.16+
Tortuosity (T)	1.16 ± 0.07	1.19 ± 0.12	**0.0318**	1.54+
Vascular velocity (VI)	0.42 ± 0.58	0.85 ± 0.64	**<0.0001**	2.63+
Vascular area (VFA)	0.82 ± 1.51	2.59 ± 3.40	**<0.0001**	2.12+

BI-RADS_US_ category (Breast Imaging Reporting and Data System for ultrasound) was a discrete feature, not a continuous numeric feature, so it is not in this table. BI-RADS_US_’s distribution is described in Section 2.2.1 BI-RADS_US_. Effect sizes are from logistic regression. A “+” indicates positive correlation with malignancy; if the + is not present, the effect of increasing the feature value is to increase the odds that the mass is benign. **Bold** indicates *p* value is significant according to the Mann-Whitney U test.

**Table 2 diagnostics-10-00631-t002:** Feature set AUC and Youden index performance with SMOTE.

Feature Set	AUC ± sErr	YI	Se at YI	Sp at YI	Sp at Se98	Sp at Se95
BI-RADS w/SMOTE	0.664 ± 0.052 **0.770 ± 0.051**	0.515 0.526	54.7 54.7	96.8 **97.9**	0.0 0.0	0.0 0.0
BI-RADS, Age w/SMOTE	0.864 ± 0.030 0.865 ± 0.031	0.608 0.607	76.6 **85.9**	**84.2** 74.7	**27.5** 9.5	42.1 **43.2**
BI-RADS, Age, CD w/SMOTE	0.891 ± 0.026 **0.901 ± 0.025**	0.661 0.661	73.4 73.4	92.6 92.6	**45.1** 26.3	47.4 **60.0**
BI-RADS, Age, GSz w/SMOTE	0.900 ± 0.024 **0.925 ± 0.022**	0.665 0.754	81.2 **85.9**	85.2 **89.5**	**48.4** 32.6	59.0 **67.4**
BI-RADS, Age, CD, GS w/SMOTE	0.934 ± 0.018 **0.958 ± 0.013**	0.780 0.811	93.8 **96.9**	84.2 84.2	44.2 **76.8**	82.1 **85.3**

AUC = area under the receiver operating characteristic curve; CD = color Doppler; GS = grayscale; YI = Youden index; sErr = standard error; Se = sensitivity; Sp = specificity; SMOTE = synthetic minority oversampling technique. **Bold** indicates higher performance between the SMOTE and non-SMOTE model values.

**Table 3 diagnostics-10-00631-t003:** Significance of paired feature set AUC performance.

Feature Set 1	Feature Set 2	*p* Value
BI-RADS_US_, Age	BI-RADS_US_, Age, GS	**0.0086** (SMOTE) **0.0116**
BI-RADS_US_, Age	BI-RADS_US_, Age, CD	**0.0320** (SMOTE) 0.1355
BI-RADS_US_, Age	BI-RADS_US_, Age, GS, CD	**0.0003** (SMOTE) **0.0006**
BI-RADS_US_, Age, GS	BI-RADS_US_, Age, CD	0.7595 (SMOTE) 0.3746
BI-RADS_US_, Age, CD	BI-RADS_US_, Age, GS, CD	**0.0104** (SMOTE) **0.0050**
BI-RADS_US_, Age, GS	BI-RADS_US_, Age, GS, CD	**0.0161** (SMOTE) **0.0352**

CD = color Doppler; GS = grayscale; SMOTE = synthetic minority oversampling technique; BI-RADS_US_ = Breast Imaging Reporting and Data System for ultrasound. Underlining indicates different features between Feature Set 2 and Feature Set 1. **Bold** indicates statistically significant *p* value.

**Table 4 diagnostics-10-00631-t004:** Performance of models at 20% drop rate with CD.

Feature Set	AUC ± sErr	95% CI	YI	Se at YI	Sp at YI
CD, 20%	**0.986 ± 0.007**	**0.947–0.999**	**0.913**	**100**	**91.3**
no CD, 20%	0.944 ± 0.022	0.889–0.977	0.817	0091.7	90.0

AUC = Area under the receiver operating characteristic curve; CD = color Doppler; GS = grayscale; CI = confidence interval; YI = Youden index; sErr = standard error; Se = sensitivity; Sp = specificity. **Bold** indicates highest performance of a metric in a column.

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
