# Peer review of "Color Doppler Ultrasound Improves Machine Learning Diagnosis of Breast Cancer"

_diagnostics, 2020, doi:10.3390/diagnostics10090631_

Round 1
Reviewer 1 Report
I have the following comments:
-The abstract should be rewritten in structured form and shortened by about 30-50% for better clarity.
-While differentiation between benign and malignant breast masses by means of color Doppler ultrasound could give valuable information before biopsy, I think it would be advisable to also assess whether CDUS-based radiomics features can allow discriminating different tumor histotypes from each other. More importantly, it would be of much clinical interest to know if such features correlate with treatment response to specific drugs, e.g. antiangiogenic agents.
Author Response
We thank the reviewer for their time and insight, and followed all suggestions as described below.
- The abstract should be rewritten in structured form and shortened by about 30-50% for better clarity.
The abstract has been structured and shortened to 200 words from 269 words, as requested. Note that although structured form is used, the structure follows the Diagnostics template, so there are no internal headings in the abstract, though the organization is (1) Background, (2) Methods, (3) Results, and (4) Conclusions.
- While differentiation between benign and malignant breast masses by means of color Doppler ultrasound could give valuable information before biopsy, I think it would be advisable to also assess whether CDUS-based radiomics features can allow discriminating different tumor histotypes from each other. More importantly, it would be of much clinical interest to know if such features correlate with treatment response to specific drugs, e.g. antiangiogenic agents.
Differentiating tumor histotypes and assessing treatment response are both important potential applications for machine learning on color Doppler features.
The 159-image database used for this study did not contain enough cases of different histotypes to support multi-class predictive modeling, but we hope that this research paves the way for future studies using color Doppler features to differentiate tumors by histotype on larger databases, as indicated in the following sentence in the manuscript’s Conclusions:
“If a larger imaging database were available, the analysis could be broken down by lesion histotype to determine which are best diagnosed by machine learning using CD.” [page 12, lines 395-396]
There has been work assessing treatment response using color Doppler features, and in fact that is what the features were originally designed for. This study uses the features for machine learning a diagnostic model of breast cancer in, but their original application has now been described in the paper, with a reference:
“Two CD features were used to measure vascularity within each manually drawn ROI: vascular fractional area (VFA) and blood flow velocity index (VI). Though the features were used in this study for machine learning, they were originally designed as measurements for assessing antivascular treatments [27].” [page 3, lines 118-121]
Reviewer 2 Report
In this study, Moustafa et coll. algorithmically extract numerical CD features from US images for machine learning, in addition to grayscale numerical features. All these computer-generated features were used along with age and the US Breast Imaging Reporting and Data System (BI-RADSUS) category to train models.
The model, based on features extracted from the vascularization of the lesions evaluated with color-doppler, aims to implement ultrasound sensitivity and specificity by acting as a second reader in the diagnosis.
The work is technically well designed, the results are well presented and commented on, the conclusions logical. However, in my opinion some additions are necessary to make it available for publication:
1. Line 40 “Ultrasound is also increasingly used as a supplemental screening tool, particularly in women 40 with dense breasts where mammographic sensitivity is limited”
Insert references, for example:
Tagliafico AS et al. A prospective comparative trial of adjunct screening with tomosynthesis or ultrasound in women with mammography-negative dense breasts (ASTOUND-2). Eur J Cancer. 2018 Nov;104:39-46.
-
Line 113: Why were only these nine GS features used? Motivate the choise
-
there are not references on the machines used. Please include this information in materials and methods
-
there are not references on the operators who performed the procedures (years of experience? Specialization? Doctors dedicated to breast care?). Please include this information in materials and methods
-
Line 296-298 “Radiomics, the diagnosis of medical images through the automated extraction and analysis of predictive quantitative features, has been increasingly used in computer tomography, magnetic resonance imaging, and positron emission tomography, but not as often in ultrasonography”. In reality there are many radiomics works in mammography and related techniques (tomosynthesis and CESM) too. To validate the results reported in this work in ultrasound it would be useful to mention in the discussion of the performance of radiomics in the different breast methods.
In this regard, I suggest some works to be included in the bibliography:
-
Fanizzi A. et al. A Machine Learning Approach on Multiscale Texture Analysis for Breast Microcalcification Diagnosis. BMC Bioinformatics 2020, 21(Suppl 2):91
-
Losurdo L. et al. Radiomics Analysis on Contrast-Enhanced Spectral Mammography Images for Breast Cancer Diagnosis:A Pilot Study. Entropy 2019, 21, 1110
-
Tagliafico AS et al. An exploratory radiomics analysis on digital breast tomosynthesis in women with mammographically negative dense breasts. Breast. 2018 Aug;40:92-96
-
Losurdo L. et al. A Gradient-Based Approach for Breast DCE-MRI Analysis.Biomed Res Int. 2018 May 16;2018:9032408
Author Response
We thank the reviewer for their time and insight and followed all of their suggestions as described below.
- Line 40 “Ultrasound is also increasingly used as a supplemental screening tool, particularly in women 40 with dense breasts where mammographic sensitivity is limited.” Insert references, for example: Tagliafico AS et al. A prospective comparative trial of adjunct screening with tomosynthesis or ultrasound in women with mammography-negative dense breasts (ASTOUND-2). Eur J Cancer. 2018 Nov;104:39-46.
Thank you, this reference has been added to the manuscript.
- Line 113: Why were only these nine GS features used? Motivate the choice.
These nine quantitative features were motivated by the conventional visual features used in the clinic to assess lesions on ultrasound, described by Stavros et al [26]. Additionally, they were the grayscale features selected as most predictive in the earlier reference studies that did not include color Doppler features, so are retained for valid comparison. The following sentence and a reference have been added to the manuscript:
“These nine features were motivated by the conventional visual features used in the clinic to assess lesions on ultrasound, described by Stavros et al [26].” [page 3, lines 111 - 112]
- There are not references on the machines used. Please include this information in materials and methods
The following has been added to the manuscript:
“The images were obtained from a Philips HDI 5000 Ultrasound System (Philips Healthcare; Best, Netherlands), operated by a registered diagnostic medical sonographer (RDMS) with over 20 years of experience.” [page 3, lines 80 - 82]
- There are not references on the operators who performed the procedures (years of experience? Specialization? Doctors dedicated to breast care?). Please include this information in materials and methods
The following has been added to the manuscript:
“The images were obtained from a Philips HDI 5000 Ultrasound System (Philips Healthcare; Best, Netherlands), operated by a registered diagnostic medical sonographer (RDMS) with over 20 years of experience.” [page 3, lines 80 - 82]
- Line 296-298 “Radiomics, the diagnosis of medical images through the automated extraction and analysis of predictive quantitative features, has been increasingly used in computer tomography, magnetic resonance imaging, and positron emission tomography, but not as often in ultrasonography”. In reality there are many radiomics works in mammography and related techniques (tomosynthesis and CESM) too. To validate the results reported in this work in ultrasound it would be useful to mention in the discussion of the performance of radiomics in the different breast methods.
In this regard, I suggest some works to be included in the bibliography:
Fanizzi A. et al. A Machine Learning Approach on Multiscale Texture Analysis for Breast Microcalcification Diagnosis. BMC Bioinformatics 2020, 21(Suppl 2):91
Losurdo L. et al. Radiomics Analysis on Contrast-Enhanced Spectral Mammography Images for Breast Cancer Diagnosis:A Pilot Study. Entropy 2019, 21, 1110
Tagliafico AS et al. An exploratory radiomics analysis on digital breast tomosynthesis in women with mammographically negative dense breasts. Breast. 2018 Aug;40:92-96
Losurdo L. et al. A Gradient-Based Approach for Breast DCE-MRI Analysis.Biomed Res Int. 2018 May 16;2018:9032408
The suggested radiomics references along with two others have been added to the manuscript.